# Iron can be microbially extracted from Lunar and Martian regolith simulants and 3D printed into tough structural materials

**Sofie M. Castelein[1]◉, Tom F. Aarts[1]◉, Juergen Schleppi[2], Ruud Hendrikx[3], Amarante J. Böttger[3], Dominik Benz[4], Maude Marechal[5], Advenit Makaya[5], Stan J. J. Brouns[1], Martin Schwentenwein[6], Anne S. Meyer[7]\*, Benjamin A. E. Lehner[1]\***

**1** Department of Bionanoscience, TU Delft, Delft, Netherlands, **2** School of Engineering and Physical Sciences, Institute for Mechanical, Process and Energy Engineering, Heriot-Watt University, Edinburgh, United Kingdom, **3** Department of Materials Science and Engineering, TU Delft, Delft, Netherlands, **4** Department of Chemical Engineering, TU Delft, Delft, Netherlands, **5** European Space Research and Technology Centre (ESTEC), ESA, Noordwijk, Netherlands, **6** Lithoz GmbH, Wien, Austria, **7** Department of Biology, University of Rochester, Rochester, New York, United States of America

◉ These authors contributed equally to this work.

\* anne@annemeyerlab.org (ASM); benjamin91lehner@gmail.com (BAEL)

**Data Availability Statement:** The data underlying the results presented in the study are available as zip file (Supporting information). All relevant data

## Abstract

*In-situ* resource utilization (ISRU) is increasingly acknowledged as an essential requirement for the construction of sustainable extra-terrestrial colonies. Even with decreasing launch costs, the ultimate goal of establishing colonies must be the usage of resources found at the destination of interest. Typical approaches towards ISRU are often constrained by the mass and energy requirements of transporting processing machineries, such as rovers and massive reactors, and the vast amount of consumables needed. Application of self-reproducing bacteria for the extraction of resources is a promising approach to reduce these pitfalls. In this work, the bacterium *Shewanella oneidensis* was used to reduce three different types of Lunar and Martian regolith simulants, allowing for the magnetic extraction of iron-rich materials. The combination of bacterial treatment and magnetic extraction resulted in a 5.8-times higher quantity of iron and 43.6% higher iron concentration compared to solely magnetic extraction. The materials were 3D printed into cylinders and the mechanical properties were tested, resulting in a 400% improvement in compressive strength in the bacterially treated samples. This work demonstrates a proof of concept for the on-demand production of construction and replacement parts in space exploration.

## Introduction

For the first time since the Apollo missions to the Moon, space agencies and commercial partners from all over the world are dedicated to send humans beyond Low Earth Orbit (LEO). The most famous examples are the European Space Agency (ESA), who have announced their plan to build an international Moon village [1]; the National Aeronautics and Space

are within the manuscript and its Supporting information files.

**Funding:** The company Lithoz provided support in the form of salaries for M. Schwentenwein and F. Ertl as well as access to their 3D printing facility, but did not have any additional role in the study design, data analysis, decision to publish, or preparation of the manuscript. The specific roles of these authors are articulated in the 'author contributions' section. This work was supported by The Netherlands Organization for Scientific Research (NWO/OCW), as part of the Frontiers of Nanoscience program. This funder had no role in study design, data collection and analysis, decision to publish, or preparation of the manuscript.

**Competing interests:** The company Lithoz provided support in the form of salaries for M. Schwentenwein and F. Ertl as well as access to their 3D printing facility. This does not alter our adherence to PLOS ONE policies on sharing data and materials. There are no patents, products in development or marketed products to declare.

Administration (NASA), who plan to send the first human to Mars [2]; and the American company SpaceX, who published their vision of realizing a human settlement on Mars [3].

A human outpost on another celestial body needs to fulfill several requirements. Transportation, shelter, supplies, waste removal, hazard protection, and power sources have to be provided to ensure its viability [4]. Every space exploration endeavor to date, however, has used resources originating from Earth, which leads to high transportation costs and, ultimately, a high degree of dependence on Earth resources. The two most common solutions to tackle the cost issue are to reduce the payload weight and to minimize the launching costs [5]. However, in the long run sustainability, earth independence, and economic feasibility can only be ensured via the direct usage of resources found in space, also called *in-situ* resource utilization (ISRU) [6].

Traditionally, research in the sector of ISRU has focused on investigating technical processes already in use on Earth and how they may be implemented in the harsh environment of space [7, 8]. When considering *in-situ* mining of building materials, direct adaptation of current mechanochemical approaches to space applications has a major disadvantage: the need for outsized factories and machinery. One novel solution to this problem is the usage of microbes to allow ISRU. These organisms can be produced in large quantities, only requiring water, a bioreactor, and an easily transportable growth medium [5]. Microbes can be successfully used in many different techniques to produce [9], extract [10], and pattern [11] material in a manner that is similar to the equivalent mechanochemical processes. Microbial production processes do not necessarily require sophisticated factories, high energy investment, or toxic chemicals [4]. Therefore, bacterial methodologies are increasingly being applied to terrestrial applications [12–15] and may be even more valuable for space exploration and colonialization [16], where resupply and a limiting initial amount of materials are major constraints [4, 17]. The research presented here aims to develop a new approach to extract iron from Martian regolith and use it in 3D printing applications (Fig 1).

The bacterium *Shewanella oneidensis*, which was shown to be feasible for space-based biomining applications [18], can reduce a variety of metal oxides under both aerobic and anaerobic conditions [19, 20]. Martian regolith contains a high concentration of iron in the form of Fe(III) [21] that can be reduced to Fe(II) by *S. oneidensis*, promoting the precipitation of magnetite which can subsequently be extracted magnetically [22]. This work has developed a new methodology for the successful extraction and utilization of iron from several different regolith types by applying *S. oneidensis*, magnetism, and additive manufacturing (Fig 1). In addition to a Martian regolith simulant (JSC-Mars1), two Lunar regolith simulants were used (EAC-1 and JSC-2A, Table 1). These two Lunar regolith simulants are chemically similar to the Martian regolith simulant and have the advantage that they have been substantially studied and standardized compared to other basaltic materials [23].

Our results indicate that none of three tested regolith simulants are toxic to *Shewanella* under either aerobic or anaerobic conditions. The combination of bacterial treatment and magnetic extraction of JSC-Mars1 regolith simulant resulted in a 5.8-fold increase in the quantity of extracted material compared to material magnetically extracted from untreated samples. Furthermore, the total iron concentration of bacterially treated and magnetically extracted JSC-Mars1 was 18.2% higher than in the starting material. The crude extracted iron was suitable for 3D printing using lithography-based ceramic manufacturing. Together, our findings provide the first proof-of-principle of bio-mediated *in-situ* iron extraction from Martian regolith simulant. Our approach is of sufficient quality to enable on-demand production of infrastructure materials as well as replacement parts for a future space colony.

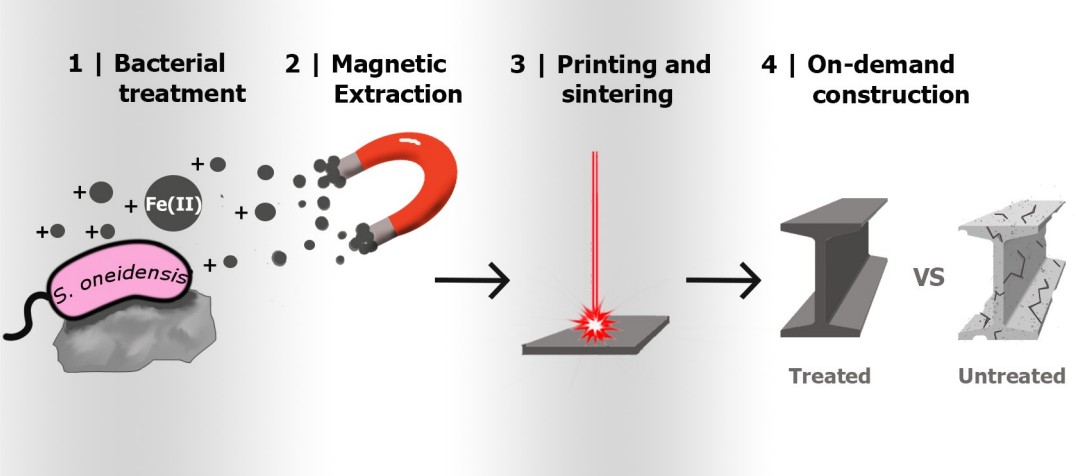

**Fig 1. Conceptual flow of the microbial biomining process.** Bacterial treatment (1) makes the material more magnetic by reducing iron. A magnet is applied to extract the magnetic material (2), and those particles are 3D printed and sintered (3). The material is then ready for construction, maintenance, and repair applications (4).

## Results & discussion

### Compositions and limitations of regolith simulants

The elemental and mineral compositions of Martian regolith simulants, in comparison to actual Martian regolith, are of critical importance to make conclusions regarding the applicability of our regolith extraction methodology on Mars. X-ray fluorescence (XRF) measurements were performed on two Lunar Mare and one Martian regolith simulant to analyze their elemental compositions. Comparison of the two Lunar Mare simulants EAC-1 and JSC-2A with the average Martian regolith values (Table 2) revealed that both contained similar oxide concentrations that resembled that of the average Martian soil. In contrast, the Martian regolith simulant JSC-Mars1 exhibited several differences in its elemental compositions compared to the average elemental composition of the actual Martian soil, especially for $SiO_2$, $Al_2O_3$, and MgO. However, the variation in Martian regolith composition due to hydrothermal activity and the differences in composition between Martian bedrock and other surface materials necessitates the use of several types of regolith with varying compositions to test for multiple different scenarios. In contrast to real Lunar regolith, the simulants also contain Fe(III) and share key properties with Martian regolith, making them excellent candidates for our experiments.

The limitations of the different simulants used in our experiments will have an impact on the interpretation of our results. Since simulants are typically sourced and produced in varying ways, and the quality-assurance processes of manufacturing companies and agencies frequently differ from each other, the actual simulant quality will vary. Both Moon regolith

**Table 1. Overview of the regolith simulants used and experiments performed in this paper.**

| Regolith simulant | Commissioned by | Simulant properties |
| --- | --- | --- |
| EAC-1 | European Space Agency | Schleppi et al. (2019) [23], Manick et al. (2014) [24] |
| JSC-2A | NASA Johnson Space Center | Schleppi et al. (2019) [23] Meurisse et al. (2018) [25] |
| JSC-Mars1 | NASA Johnson Space Center | Allen et al. (1998) [26] |

**Table 2. Average major metal oxide composition (wt%) at different Mars locations [29–31] and the composition of three different regolith simulant types obtained via X-ray fluorescence (XRF) as well as reported by the suppliers (JSC-Mars1 [32], JSC-2 [23], EAC-1 [23]).**

| Metal oxides (wt%) | Mars Viking 1 [29] | Mars Viking 2 [29] | Mars Pathfinder [30] | JSC-Mars 1 (Supplier) [32] | JSC- Mars1 (XRF) | JSC-2A (Supplier) [23] | JSC-2A (XRF) | EAC-1 (Supplier) [23] | EAC-1 (XRF) |
|---|---|---|---|---|---|---|---|---|---|
| $SiO_2$ | 43 | 43 | 44 | 43.5 | 42.19 | 47.5 | 44.89 | 43.7 | 43.58 |
| $Al_2O_3$ | 7.3 | 7 | 7.5 | 23.3 | 23.48 | 15 | 19.61 | 12.6 | 11.45 |
| $Fe_2O_3$ | 18.5 | 17.8 | 16.5 | 15.6 | 17.55 | 10.75 | 12.78 | 12 | 12.66 |
| MgO | 6 | 6 | 7 | 3.4 | 3.42 | 9 | 4.72 | 11.9 | 14.08 |
| CaO | 5.9 | 5.7 | 5.6 | 6.2 | 6.01 | 11 | 10.1 | 10.8 | 10.18 |
| $K_2O$ | <0.15 | <0.15 | 0.3 | 0.6 | 0.49 | 0.8 | 0.67 | 1.3 | 1.18 |
| $TiO_2$ | 0.66 | 0.56 | 1.1 | 3.8 | 3.80 | 2 | 2.23 | 2.4 | 2.15 |

simulants JSC-1A and EAC-1 have been mined from the surface of a volcanic basaltic deposit. However, JSC-1A was mined from a volcanic ash deposit in the San Francisco volcano field in Arizona (35˚20' N, 111˚17' W), which erupted only approximately 0.15 ± 0.03 million years ago [27], whereas EAC-1 was mined from a deposit in the Eifel region (50˚41'N, 7˚19'E), which is approximately 20 million years old. Although both simulants could have been in contact with water, air, and vegetation prior to being mined, the older age of EAC-1 increases the chance of exposure to processes that could change its composition. For example, EAC-1 shows alterations such as the presence of chlorite, which is usually found in igneous rocks and results from water interacting with pyroxene minerals. In contrast to EAC-1, JSC-2A, which is the successor of JSC-1A, is produced from synthetic minerals [25] and therefore shares the composition of JSC-1A without its environmental alterations due to air, water and vegetation. These differences in composition caused by varying length of exposure time to the elements are valuable for our research since compositional differences are also observed in the Martian bedrock. Furthermore, evaluation of a variety of regolith types can give insights into the most suitable regolith type for our method.

The Martian regolith simulant JSC-Mars1 was sourced on the island of Hawaii from sieved palagonatized tephra ash from the 1843 Mauna Loa lava flow on Pu'u Nene (19˚41'N, 155˚29'W) [28]. Although all three regolith types were excavated close to the surface of the Earth, JSC-Mars1, at 175 years of age, is very young compared to the Lunar simulants EAC-1 and JSC-1A. Martian regolith contains sulfate (avg. 6.16%, detected as sulfur trioxide) and chlorine (avg. 0.68%), neither of which have been detected in our regolith samples by the suppliers or in the work presented here. The iron content of the Viking and Pathfinder Mars samples are best approximated by JSC-Mars1, while the aluminum content is closer to that of EAC-1 and the magnesium content is the closest to JSC-2A (Table 2). These resemblances to real Martian regolith make all three regolith types (JSC-Mars1, EAC-1, and JSC-2A) good targets to study the interaction with *S. oneidensis*.

Equally as important as the elemental composition of regolith samples is the phase composition, which determines their accessibility for biological treatment with *S. oneidensis*. The mineral composition of the different regolith simulants was inspected by X-ray diffraction (XRD) analysis. The most abundant Martian minerals plagioclase, olivine, and pyroxene were detected in all three regolith simulants. EAC-1 showed the presence of nepheline and manganese iron oxide, both of which were not yet detected on Mars, and JSC-2A contained a small fraction of magnetite, similar to the results from Mars Curiosity (Table 3).

It should be taken into account that the regolith simulants used in this study are rich in amorphous phases, while actual Martian soil consists of an approximately equal distribution of amorphous and crystalline phases [25, 33]. The higher amount of amorphous phases

**Table 3. X-ray diffraction analysis of the Mars Rocknest soil [21] and the three different regolith simulants.**

| Mineral | IUPAC nomenclature | Mars Curiosity Rocknest soil [21] | JSC-Mars1 (XRD) | JSC-2A (XRD) | EAC-1 (XRD) |
|---|---|---|---|---|---|
| Plagioclase (Anorthite) | $(Ca_{0.57(13)}Na_{0.43})(Al_{1.57}Si_{2.43})O_8$ | 40.8% | + | + | + |
| Olivine (Forsterite) | $(Mg_{0.62(3)}Fe_{0.38})_2SiO_4$ | 22.4% | + | + | + |
| Pyroxene (Augite) | $(Mg_{0.88(10)}Fe_{0.37}Ca_{0.75(4)})Si_2O_6$ | 14.6% | + | (+)* | + |
| Pyroxene (Pigeonite) | $(Mg_{1.13}Fe_{0.68}Ca_{0.19})Si_2O_6$ | 13.8% | - | - | - |
| Magnetite | $Fe_3O_4$ | 2.1% | - | (+)* | - |
| Anhydrite | $CaSO_4$ | 1.5% | - | - | - |
| Quartz | $SiO_4$ | 1.4% | - | - | - |
| Manganese Iron Oxide | $Mn_{0.43}Fe_{2.57}O_4$ | - | - | - | + |
| Nepheline | $(K_{0.69}Na_{3.03}Ca_{0.03}Fe_{0.04}Al_{13.75}Si_{0.21})(SiO_4)_4$ | - | - | - | + |

Minerals that were detected are denoted with "+"; minerals that were not detected are denoted with "-".

* These elements were only detected in the magnetically extracted fraction of bacterially treated and bacterially untreated material.

influences the magnetic properties of the regolith simulants used. For example, it has been reported that JSC-Mars1 is more magnetic than the actual Martian soil [26], which could affect the amount of magnetic material extracted in our approach. Our method, however, is aimed at quantifying the increase in the amount of magnetically extractable material upon bacterial treatment, rather than determining the exact amount of material we can extract. To study whether regolith simulants containing more crystalline phases are also compatible with our approach, and to potentially better determine the amount of material that can be extracted in future biomining approaches, it would be worthwhile to perform follow-up studies using the recently developed regolith simulant MGS-1 [33].

Comparison of all three potential regolith types with the Martian regolith measurements revealed strong similarities in both mineral and elemental composition between the regolith simulants and the Martian target regolith. The Martian and Lunar regolith simulants are therefore suitable to use for our bio-based method.

## Toxicity of regolith simulants in aerobic and anaerobic conditions

To acquire a working concentration of regolith for our bacterial treatment, the toxicity of the Lunar and Martian regolith simulants for *S. oneidensis* was determined at a range of regolith concentrations under the presence (aerobic, Fig 2A, 2C and 2E) and absence (anaerobic, Fig 2B, 2D and 2F) of oxygen. Colony forming units (CFU) and optical density (O.D.$_{600}$) were measured to examine the effect of increasing regolith simulant concentrations and different regolith types on both bacterial viability and growth. All control samples, which contained regolith but no bacterial cells, contained 0 CFU/μL at all time points. The optical density of control samples containing regolith but no *S. oneidensis* was subtracted from each data point.

*S. oneidensis* showed little or no reduction in maximal CFU concentration or O.D.$_{600}$ absorbance upon exposure to increasing concentrations of the different regolith types. The exception to this trend was a decrease in O.D.$_{600}$ absorbance that was observed upon exposure to increasing regolith concentrations in bacteria exposed to JSC-Mars1 compared to the no regolith control. This result may reflect the difficulty in obtaining reliable absorbance measurements in the presence of high Fe(III) concentrations due to the optical properties of Fe(III) [34], as this trend was not observed in the corresponding CFU measurements. The bacteria exposed to JSC-2A and EAC-1 under aerobic conditions demonstrated higher maximal CFU concentrations compared to anaerobic conditions, indicating that the aerobic conditions resulted in a better growth environment. In contrast, bacteria exposed to JSC-Mars1 had a

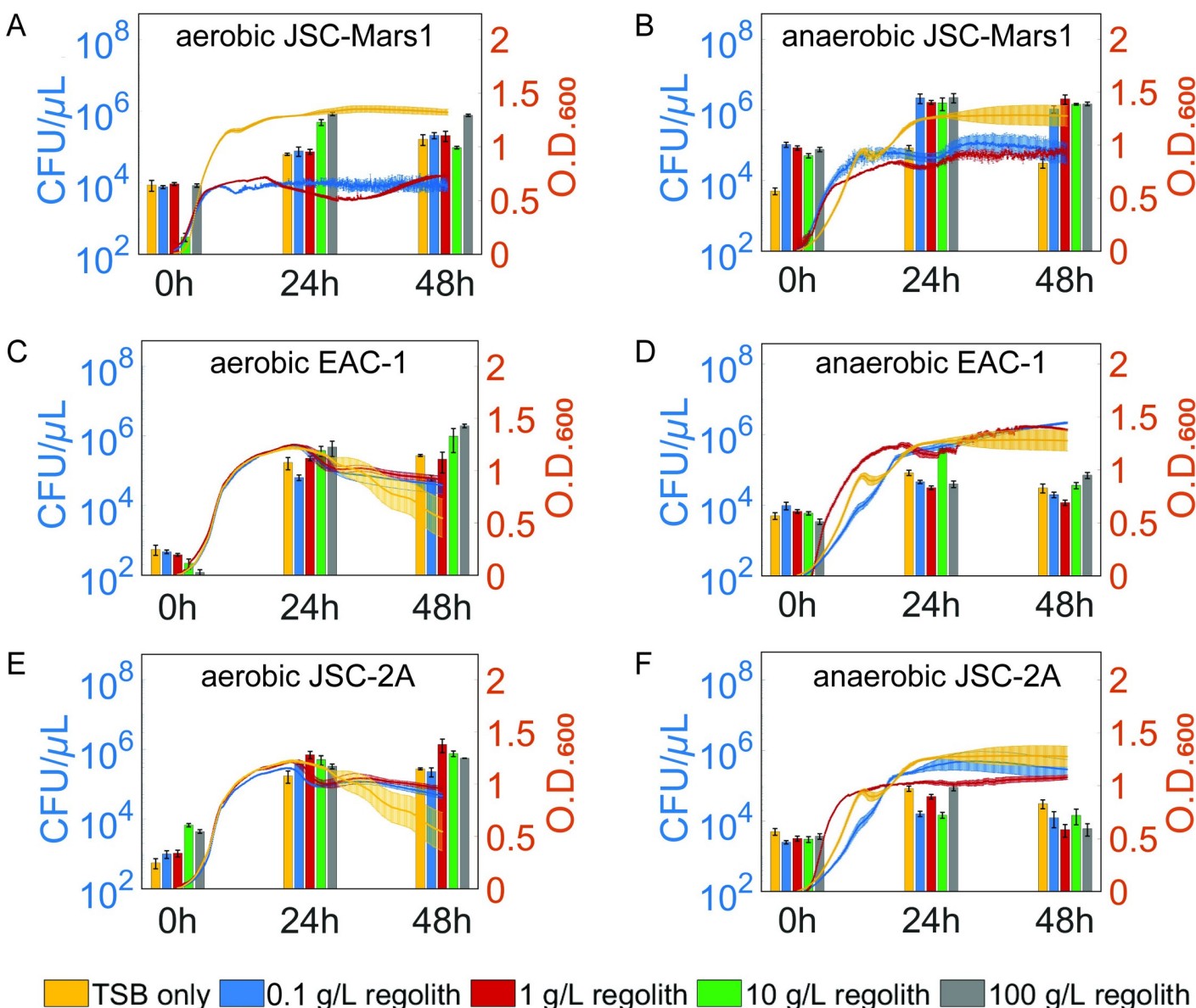

**Fig 2. Evaluation of the toxicity of different Lunar and Martian regolith simulants to *S. oneidensis* under aerobic and anaerobic conditions.** Colony forming units (CFU/μL) are depicted on the left axis and the O.D.$_{600}$ measurements on the right axis. The yellow, blue, red, green, and grey bars represent regolith concentrations of 0, 0.1, 1, 10, and 100 g/L respectively. Optical density at 600 nm of control samples without *S. oneidensis* were subtracted from the O.D.$_{600}$ measurements. The error bars represent the standard error of the mean. Aerobic growth behavior of *Shewanella oneidensis* was measured in the presence of JSC-Mars1 (A), EAC-1 (C) and JSC-2A (E), and anaerobic growth of *Shewanella oneidensis* was measured in the presence of JSC-Mars1 (B), EAC-1 (D) and JSC-2A (F).

higher maximal CFU upon anaerobic growth, possibly due to the higher amount of Fe(III) that is available as a terminal electron acceptor in this regolith type. The starting CFUs of all anaerobic samples was slightly higher due to time required to transfer the samples from the anaerobic glove box to agar plates. Overall, the O.D.$_{600}$ and CFU results indicate similar growth behavior for bacteria exposed to different concentrations and types of regolith under the same environmental conditions. These results suggest low toxicity or affinity for any particular type or concentration of regolith towards *S. oneidensis*, with the exception of an affinity for JSC-Mars1 under anaerobic growth conditions.

## Magnetic extraction and Fe(II) content upon aerobic bacterial treatment

Construction material will be of utmost importance for any space colony, and magnetism would be a simple way to separate different ores. *S. oneidensis* can reduce Fe(III) to Fe(II) at its cell membrane together with a local pH increase, allowing for the precipitation of magnetite, a ferrimagnetic iron oxide. We hypothesized that this methodology might be applied to increase the quality and quantity of extractable magnetic material from regolith simulants. In order to produce more magnetic minerals within regolith and therefore concentrate the iron content, *S. oneidensis* was incubated an extended period of 168 hours with 10 g/L of regolith simulant.

Each bacterially treated or untreated regolith simulant sample was assayed to determine both the change in amount of magnetic material and in aqueous Fe(II) concentration during treatment. To assess the amount of magnetic material at different timepoints, both with and without bacterial treatment, an aerobic small-scale magnetic extraction experiment was performed utilizing handhold neodymium magnets (Fig 3A).

To determine the aqueous iron(II) concentration ($Fe(II)_{(aq)}$) with and without bacterial treatment at different timepoints, a colorimetric assay was performed using 1,10-Phenanthroline as a complexing reagent for Fe(II) (Fig 3B) [35]. An increase in Fe(II) bound to the complexing reagent can be measured via absorbance at $O.D._{510}$. NaF was also added to complex Fe(III), thereby preventing interference with the absorbance measurement. This assay was performed on samples with a known aqueous iron concentration to generate a standard curve for $Fe(II)_{(aq)}$ and to test for any potential interference by $Fe(III)_{(aq)}$ (Fig 3B). A linear fit to a Fe(II) standard curve showed a good fit to a linear regression model ($R^2 = 0.9995$). No significant difference was observed between the two curves ($y_{Fe(II)+Fe(III)} = 4.3e-3^*x + 8.0e-2$; $y_{Fe(II)} = 4.4e-3^*x + 8.4e-2$, One-way ANOVA, p = 0.99), indicating that no interference in the Fe(II) absorbance measurements was caused by the presence of up to 100 ppm Fe(III).

The amount of magnetically extracted material from EAC-1 increased significantly (Fig 3C) from $1.4 \pm 0.44$ mg in the 0h sample to $2.0 \pm 0.19$ mg (1.4-fold) after 168 hours of incubation with *S. oneidensis* ($p_{S0h-S168h} = 0.044$), while there was no significant difference between the no-bacteria control at 0h and at 168h (One-way ANOVA with Tukey PostHoc test, n = 6, $p_{C0h-S0h} = 0.83$, $p_{C0h-C168h} = 0.35$). The 168h-incubated bacterially treated sample also contained significantly higher amounts of magnetic material than the 168h no-bacteria control ($p_{S168h-C168h} = 0.0069$).

Upon determination of the Fe(II) concentration of bacterially treated samples of EAC-1, the no-bacteria controls did not display any significant differences between the timepoints (Fig 3D) (One-way ANOVA with Tukey PostHoc test, n = 9, p > 0.05). The absorbance of the samples containing bacteria increased significantly within the first 48 hours ($p_{S0h-S48h} = 0.0015$) from $0.33 \pm 0.029$ A.U. to $0.40 \pm 0.023$ A.U., but was not significantly different from the control at 48h ($p_{C48h-S48h} = 0.87$). The Fe(II) concentration of the bacterially treated samples continued to increase throughout the remainder of the experiment to a final $O.D._{510}$ of $0.61 \pm 0.15$ A.U., a 1.9-fold increase ($p_{S0h-S168h} = 1e-8$), corresponding to an aqueous iron concentration ($Fe(II)_{(aq)}$) of $132.5 \pm 17.5$ ppm. The absorbance in the bacterial sample after 168 hours was also significantly greater than the control at 168h ($p_{C168h-S168h} = 0.004$), indicating a successful and significant reduction of Fe(III) to Fe(II) in EAC-1 after 72 and 168 hours of bacterial treatment.

In comparison to EAC-1, the amount of magnetically extracted material from JSC-2A regolith (Fig 3E) increased more strongly between the 0h ($1.5 \pm 0.21$ mg) and 168h ($2.5 \pm 0.27$ mg) bacterially treated sample ($p_{S0h-S168h} = 0.038$). The 168h bacterially treated sample also contained significantly higher amounts of magnetically extracted material than the 0h ($p_{C0h-S168h} = 0.019$), and the 168h no-bacteria control ($p_{S168h-C168h} = 0.0027$). No significant differences

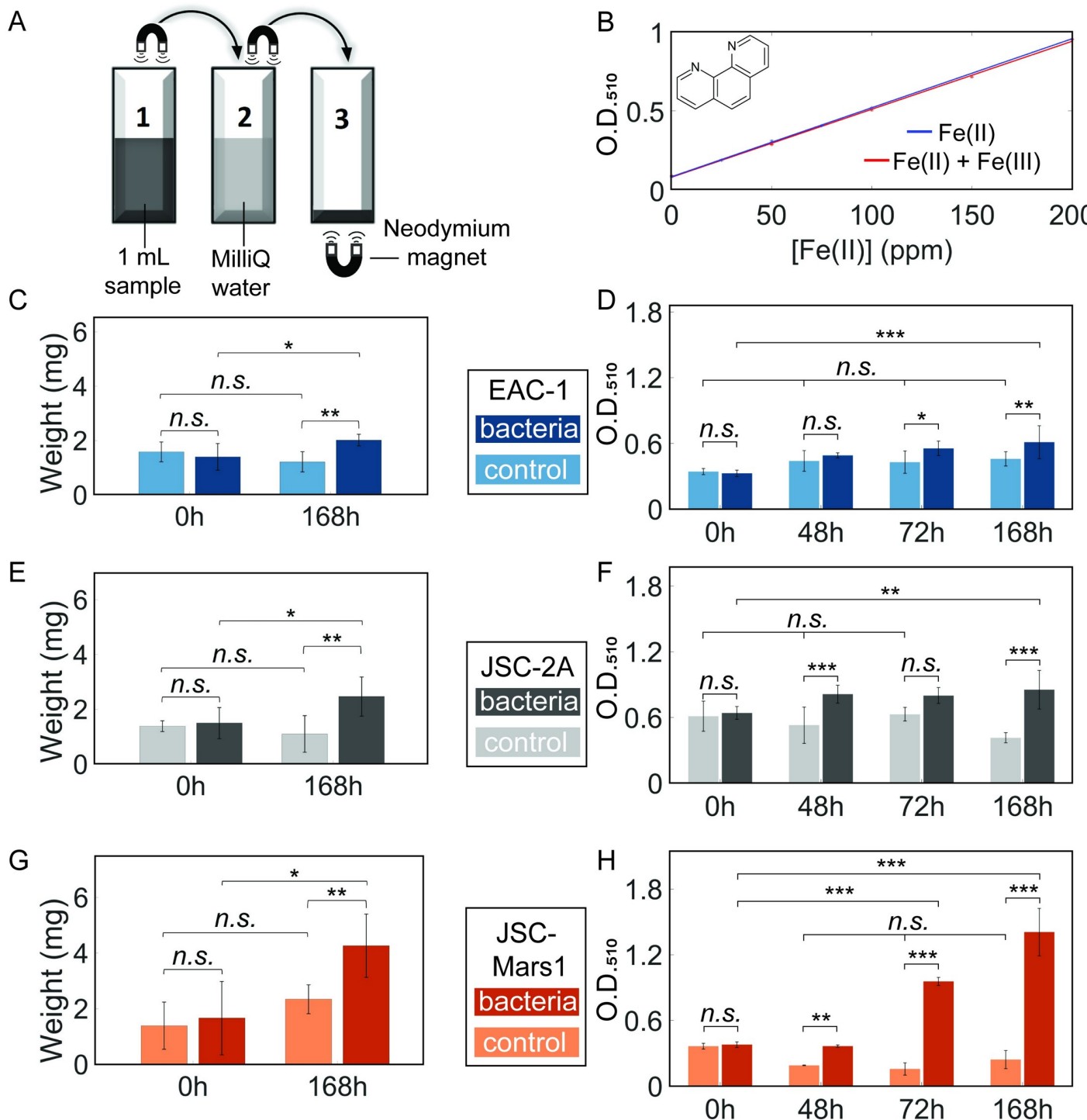

**Fig 3. Aerobic extraction of magnetic material and determination of aqueous ferrous iron concentration from EAC-1 (blue), JSC-2A (grey), and JSC-Mars1 (orange) regolith simulants.** The darker color in each plot represents the bacterial sample, the lighter one the non-bacterial control. (A) The set-up for small-scale magnetic extraction. In short, 1 mL sample (1) was pipetted into a cuvette, then the magnetic fraction was extracted via a magnet, washed in a MilliQ water (2), and finally dried in a previously weighed cuvette (3). The weight difference of the cuvette before and after this treatment represents the extracted magnetic material. (B) Absorbance of Fe(II) iron standards bound to 1,10-Phenanthroline in the absence and presence of Fe(III). O.D.$_{510}$ was measured for a range of 0–200 ppm Fe(II)$_{(aq)}$ with or without a constant concentration of 100 ppm Fe(III)$_{(aq)}$. (C, E, G) The weight of magnetically extracted material from samples containing 10g/L EAC-1 (C), JSC-2A (E), or JSC-Mars1 (G) solution with and without *S. oneidensis* after 0h and 168h. An increase of the magnetically extractable material was measured for all of the bacterially treated samples (n = 6). (D, F, H) Absorbance of the colorimetric iron determination of 10 g/L EAC-1 (D), JSC-2A (F), and JSC-Mars1 (H) treated with

aerobic *S. oneidensis* over 168 hours. A consistent increase of the $Fe(II)_{(aq)}$ concentration was measured for all of the bacterially treated samples (n = 9). Mean plus standard deviation is shown for the bar plots (One-way ANOVA with Tukey PostHoc test: N.s. > 0.05; * ≤ 0.05; ** ≤ 0.01; *** ≤ 0.001).

were detected between the no-bacteria controls at 0h and 168h ($p_{C0h-C168h}$ = 0.83), nor between the bacterial sample and the control at 0h (n = 6, one-way ANOVA with Tukey Post-Hoc test, $p_{S0h-C0h}$ = 0.98). *S. oneidensis* was, therefore, able to increase the amount of magnetically extracted material in JSC-2A significantly (1.7-fold).

The starting $Fe(II)_{(aq)}$ concentration in the JSC-2A bacterial samples (Fig 3F) at 0h was double (0.64 ± 0.059 A.U.) that of EAC-1. The Fe(II) concentration in the bacterially treated sample was significantly higher than in the no bacteria control at both 48h ($p_{C48h-S48h}$ = 3e-5) and 168h ($p_{S168h-C168h}$ = 1e-8), but not at 72h ($p_{C72h-S72h}$ = 0.087). A significant increase in Fe(II) was observed between the bacterial sample at 0h and 48h ($p_{S0h-S48h}$ = 0.035) as well 168h ($p_{S0h-S168h}$ = 0.0051) to a maximum of 0.85 ± 0.18 A.U. (1.3-fold), corresponding to a $Fe(II)_{(aq)}$ concentration of 192.5 ± 25 ppm. However, the control at 168h was also significantly different from the control at 0h ($p_{C0h-C168h}$ = 0.0086). The rest of the controls did not have any significant differences between each other at any timepoint (n = 9, one-way ANOVA with Tukey PostHoc test, p > 0.05). These results show that a significant reduction of Fe(III) to Fe(II) by *S. oneidensis* can also happen in the bacterially treated JSC-2A regolith simulant, but the increase in Fe(II) was smaller since JSC-2A contains less Fe(III) than EAC-1.

The JSC-Mars1 sample (Fig 3G) showed the highest increase in magnetically extractable material (2.5-fold) between the 0h (1.7 ± 1.2 mg) and the 168h (4.3 ± 1.04 mg; $p_{S0h-S168h}$ = 5.5e-05) bacterial sample among all regolith simulants tested. The magnetically extracted weight of the 168h bacterial sample was also significantly higher than the 168h control ($p_{S168h-C168h}$ = 0.025) and the 0h control ($p_{C0h-S168h}$ = 1.02e-5). There were no significant differences between the controls ($p_{C0h-C168h}$ = 0.32), nor between the control and bacterial sample at 0h (One-way ANOVA with Tukey PostHoc test, $p_{S0h-C0h}$ = 0.89).

The aqueous ferrous iron ($Fe(II)_{(aq)}$) concentration of JSC-Mars1 regolith (Fig 3H) was significantly higher in the 0h no-bacteria control compared to the 48h and 72h controls (p < 0.05), but not compared to the 168h control ($p_{C0h-C168h}$ = 0.071). However, significant changes in the $Fe(II)_{(aq)}$ concentration occurred in the bacterial JSC-Mars1 sample, which increased from an initial value of 0.38 ± 0.025 A.U. at 0h to 0.95 ± 0.036 A.U. after 72h ($p_{S0h-S72h}$ = 1e-8) and to 1.4 ± 0.22 A.U. after 168h equaling a 3.7-fold increase and a total $Fe(II)_{(aq)}$ concentration of 330 ± 35 ppm ($p_{S0h-S168h}$ = 1e-8). For all timepoints besides 0h, the bacterial sample was also significantly different from its respective controls (One-way ANOVA with Tukey PostHoc test, p < 0.05). JSC-Mars1 was therefore the best candidate for bacterial treatment and magnetic extraction, showing the highest increase in extractable material and dissolved $Fe(II)_{(aq)}$ concentration. This regolith simulant was therefore explored further to test the effect of anaerobic incubation with *S. oneidensis* on Fe(II) content and extraction of magnetic material.

## Magnetic extraction and $Fe(II)_{(aq)}$ content upon anaerobic bacterial treatment

Anaerobic growth experiments were performed to assess the reduction capability of *S. oneidensis* without the availability of oxygen as additional electron acceptor. The first set of anaerobic experiments used rich TSB as a growth medium. Quantification of the magnetically extracted materials (Fig 4A) displayed a significant increase after both 72h (4.0 ± 0.74 mg) and 168h (3.6 ± 1.4 mg) compared to the no-bacteria control samples and the 0h bacterial sample

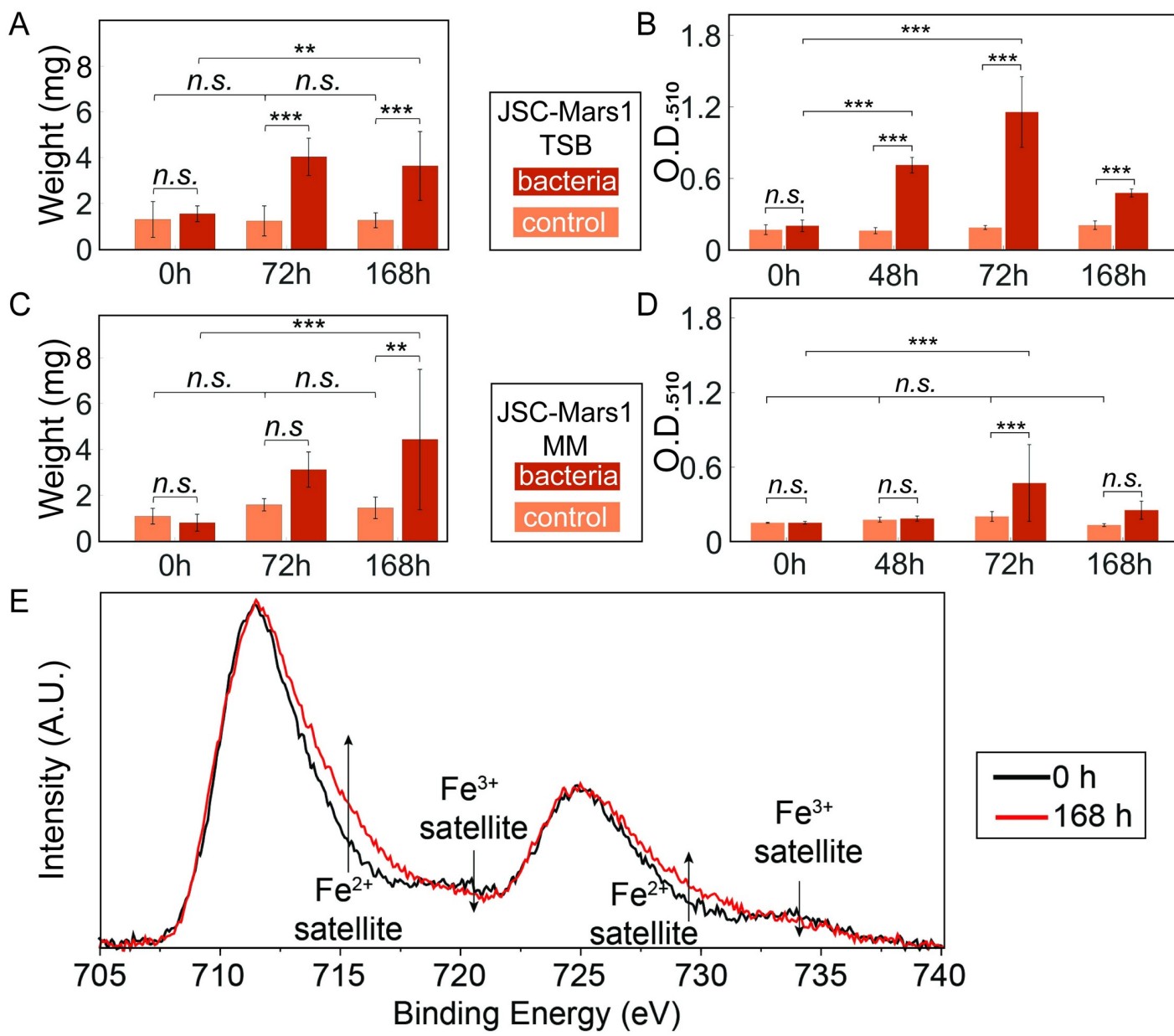

**Fig 4. Anaerobic extraction of magnetic material from bacterially treated JSC-Mars1 regolith simulant.** The amount of magnetically extracted material from a 10g/L JSC-Mars1 regolith simulant anaerobically incubated with (red) or without (orange) *Shewanella* bacteria in TSB medium (A) or defined minimal medium (MM) (C) after 0h, 72h, and 168h. The O.D.$_{510}$ values of the colorimetric assay to determine the Fe(II)$_{(aq)}$ concentration of 10 g/L JSC-Mars1 regolith anaerobically incubated with (red) or without (orange) *Shewanella* bacteria in TSB medium (B) and defined minimal medium (D). The standard deviation is given for all bar plots (One-way ANOVA with Tukey PostHoc test: N.s. > 0.05; $^{*} \leq 0.05$; $^{**} \leq 0.01$; $^{***} \leq 0.001$). (E) X-ray photoelectron spectroscopy of iron in JSC-Mars1 regolith samples. A change is visible in the Fe(II) and Fe(III) satellite peaks between the 0h (black line) and the 168h (red line) timepoints of incubation with *S. oneidensis*.

(1.5 ± 0.31 mg) ($p_{S72h-C72h}$ = 3.3e-5, $p_{S0h-S72h}$ = 0.00021, $p_{S168h-C168h}$ = 0.0059, $p_{S0h-S168h}$ = 0.00061). The amount of material extracted from the bacterial samples increased 2.7-fold between 0h and 72h and 2.4-fold between 0h and 168h. No significant increase in extracted material was observed in the no-bacteria sample over time, nor between the bacterially treated samples at 72h and 168h (One-way ANOVA with Tukey PostHoc test, p > 0.05). Therefore, the amount of magnetically extracted material for JSC-Mars1 was similar after 168h for both

the anaerobic and aerobic conditions, but unlike the aerobic experiments, the maximal yield was already reached after 72 hours upon anaerobic treatment.

The colorimetric assay to test the aqueous ferrous iron concentration ($Fe(II)_{(aq)}$) displayed no significant differences between the absorbances of the control samples at any time point (Fig 4B), nor between the bacterial sample and the control at 0h (One-way ANOVA with Tukey PostHoc test, $p > 0.05$). The bacterially treated sample showed a 3.6-fold, significant increase of $Fe(II)_{(aq)}$ after 48h ($0.71 \pm 0.093$ A.U.) and a 5.8-fold significant increase after 72h ($1.16 \pm 0.29$ A.U.) compared to the 0h sample ($0.20 \pm 0.024$ A.U.) ($p_{S48h-C48h}$ = 1e-8, $p_{S0h-S48h}$ = 1e-8, $p_{S72h-C72h}$ = 1e-8, $p_{S0h-S72h}$ = 1e-8). Interestingly, the $Fe(II)_{(aq)}$ concentration decreased significantly after 168h ($0.48 \pm 0.032$ A.U.) compared to the 72h bacterial sample (One-way ANOVA with Tukey PostHoc test, $p_{S168h-S72h}$ = 1e-8). The greater yield and faster increase of $Fe(II)_{(aq)}$ in the anaerobic JSC-Mars1 samples compared to the aerobic conditions indicates that the reduction of Fe(III) to Fe(II) was improved under anaerobic conditions.

JSC-Mars1 regolith simulant was also incubated anaerobically with *S. oneidensis* in a minimal, defined medium instead of the nutrient-rich TSB medium. A minimal medium holds the advantage that its nutrients can be fully utilized, which is essential to reduce the transportation weight for space applications. Quantification of the magnetically extracted material (Fig 4C) showed no significant differences between any of the time points for the no-bacteria controls ($p > 0.5$). However, the bacterially treated, magnetically extracted material increased significantly after 72h to $3.1 \pm 0.70$ mg (3.8-fold) and 168h to $4.4 \pm 2.8$ mg (5.5-fold) in comparison to the 0h bacterially treated sample ($0.8 \pm 0.34$ mg) and the no-bacteria control at 168h ($p_{S0h-S72h}$ = 0.05, $p_{S168h-C168h}$ = 0.0059, $p_{S0h-S168h}$ = 0.00061). Unlike the anaerobic sample with TSB, no significant difference between the bacterially treated sample and the control after 72h was measured (One-way ANOVA with Tukey PostHoc test, $p_{S72h-C72h}$ = 0.36). The absolute weight of the magnetically extracted material after 168h of anaerobic bacterial treatment of JSC-Mars1 in minimal defined medium ($4.4 \pm 2.8$ mg) was comparable to the weight obtained from the anaerobic extractions in TSB ($3.6 \pm 1.4$ mg) and the aerobic JSC-Mars1 treatment ($4.3 \pm 1.04$ mg).

The $Fe(II)_{(aq)}$ concentration in the colorimetric assay did not display significant differences between the no-bacteria controls at any timepoint (Fig 4D), nor between the control and the bacterial sample at 0h ($p > 0.05$). A significant, 3.1-fold increase in Fe(II) to a maximum concentration of $97.5 \pm 0.47$ ppm was observed for the bacterial sample after 72h compared to 0h ($p_{S0h-S72h}$ = 0.0082), followed by a significant decrease to $45 \pm 1.75$ ppm after 168h ($p_{S72h-S168h}$ = 0.013). Only the 72h bacterially treated sample showed a significantly higher $Fe(II)_{(aq)}$ concentration compared to its respective 72h no bacteria control (One-way ANOVA with Tukey PostHoc test, $p_{S72h-C72h}$ = 0.00089). However, the increase in ferrous iron was lower than observed in the experiments using TSB. The experiments in a minimal defined medium showed a significant increase in both the $Fe(II)_{(aq)}$ concentration as well as the magnetically extracted material after 72h and required the least resources, which makes this condition the preferred choice for space exploration.

We performed high-resolution X-ray photoelectron spectroscopy (XPS) measurements (Fig 4E) to investigate the bacterial reduction of Fe(III) to Fe(II) in the anaerobically bacterially treated JSC-Mars1 samples after 0 and 168 hours. An increase in the XPS signal was observed in the 168-hour sample at approximately 715 eV and 728 eV, corresponding to Fe(II) satellite peaks. Additionally, the signals at 720 eV and 734 eV, corresponding to Fe(III) satellite peaks, showed a lower intensity for the 168h treated samples. Therefore, the XPS measurement confirmed the results of the colorimetric assay that anaerobic treatment of JSC-Mars1 regolith with *S. oneidensis* resulted in a successful reduction of Fe(III) to Fe(II).

## Iron and silicon concentration in the treated and untreated regolith simulants

A higher iron oxide and lower silicon oxide concentration are critical to improve the electric as well as mechanical properties of the regolith and decrease the melting point of the material [36]. We postulate that the magnetic and bacterial treatment will concentrate iron and therefore improve the mechanical properties of the treated sample. To analyze the weight percent of iron oxides (Fig 5A) and silicon oxides (Fig 5B) in treated regolith samples as well as untreated regolith samples, to which no magnetic or bacterial methodology were applied, X-ray fluorescence measurements (XRF) were performed. To obtain the amount of material required for XRF experiments, large-scale magnetic extractions were performed (details in "Materials & methods" section). The XRF data for our experimental samples can be directly compared to the values given by the supplier. The untreated regolith samples were found to have iron and silicon weight percentages equivalent to the values reported by the suppliers. In all three regolith types, an increase in the iron wt% (EAC-1 43.6%, JSC-2A 17.1%, JSC-Mars1 18.2%) and a decrease in the silicon wt% (EAC-1–22.6%, JSC-2A -7.4%, JSC-Mars1–25.0%) were observed upon bacterial treatment and magnetic extraction compared to the untreated material. This alteration of the composition upon treatment indicates that not only the quantity but also the quality of extracted material for construction applications was improved through the application of *S. oneidensis* and magnetic extraction.

## Ultimate compressive strength of 3D prints

To test whether bacterially and magnetically extracted regolith could produce a structural material with improved mechanical properties, 3D prints of treated and untreated JSC-2A were produced via lithography-based additive manufacturing followed by sintering (Fig 6).

For the untreated samples, an average ultimate compressive strength of 3.33 ± 0.39 MPa was measured (n = 9). Strikingly, for the treated samples, an average ultimate compressive strength of 13.18 ± 3.49 MPa was found (n = 5), indicating a significant 396 ± 115% increase of the ultimate compressive strength after bacterial treatment of the material (One-way ANOVA with Tukey PostHoc test, p = 1.7e-6). These results indicate that the sintered 3D prints produced from treated regolith are stronger than the ones produced from untreated regolith. The large standard deviation of the increase in ultimate compressive strength was likely due to physical distortions observed in the treated samples, causing a variation in their structural composition. The amount of magnetically extracted treated material was limited, and to avoid loss of material by adhesion on the milling balls, the milling on the treated material was limited, compared to the untreated material. This led to coarser particles being fed into the additive manufacturing process. To address this less homogenous feedstock, the sintering temperature was increased to 1100˚C, compared to that of the untreated material (1050˚C). Nevertheless, this coarser powder resulted in a different rheological behavior of the powder-binder suspension during additive manufacturing, causing uneven spreading of the powder-binder suspension and the onset of macroscopic defects in the printed material. The resulting defects affected the crack propagation and therefore the homogeneity of the compressive test results. Processing more material would lead to a finer powder, which could improve at the same time the geometry of the samples (no distortion causing stress-concentrations) and reduce the amount and size of the defects. As a result, increased Ultimate Compressive Strength values would be expected. Albeit inhomogeneous, the results reported here can therefore be considered as a conservative estimate of the compressive strength of treated 3D printed material. The bacterial and magnetic treatment is considered to have the potential for higher increase in strength than reported here.

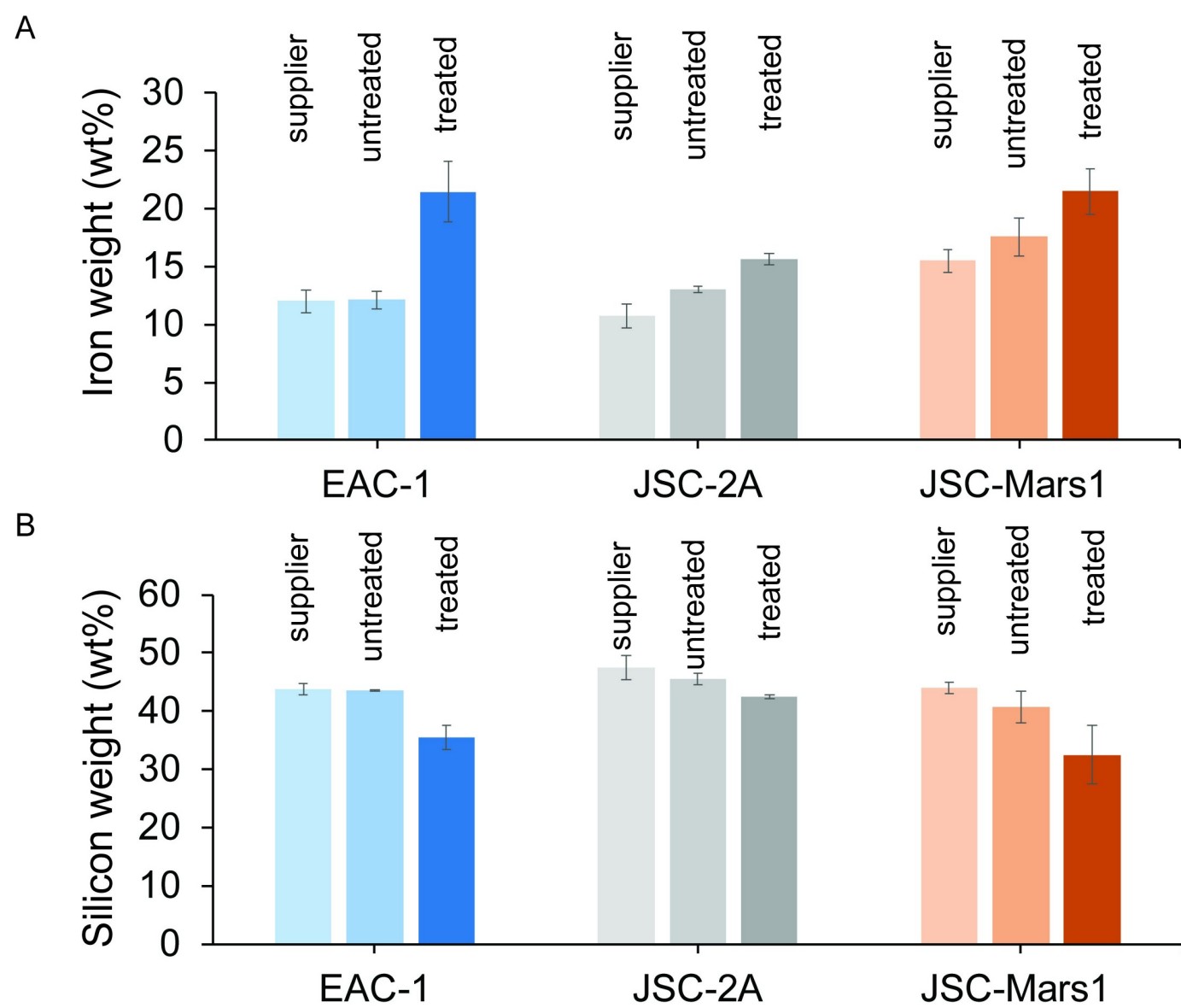

**Fig 5.** X-ray fluorescence spectroscopy analyzing the (A) iron concentration (wt%) and (B) silicon concentration (wt%) of the three different regolith simulants as given by the supplier ("supplier", shown in bright colors); measured without treatment (untreated, in intermediate colors); or bacterially treated and magnetically extracted (treated, in dark colors). The different regolith types are indicated in blue for EAC-1, grey for JSC-2A, and red for JSC-Mars1. Error bars display the standard deviation.

## Conclusion

In this work, we demonstrate an increase in the iron concentration and the quantity of magnetically extracted material upon bacterial treatment of three different regolith simulants. This increased iron concentration and quantity are both critical for applying our methodology to 3D printing applications, demonstrating an on-demand production work-flow for a space colony.

*S. oneidensis* was chosen because of its capacity to reduce Fe(III) to Fe(II) in a range of different environments, its reductive bioleaching activity of minerals [37], and its ability to

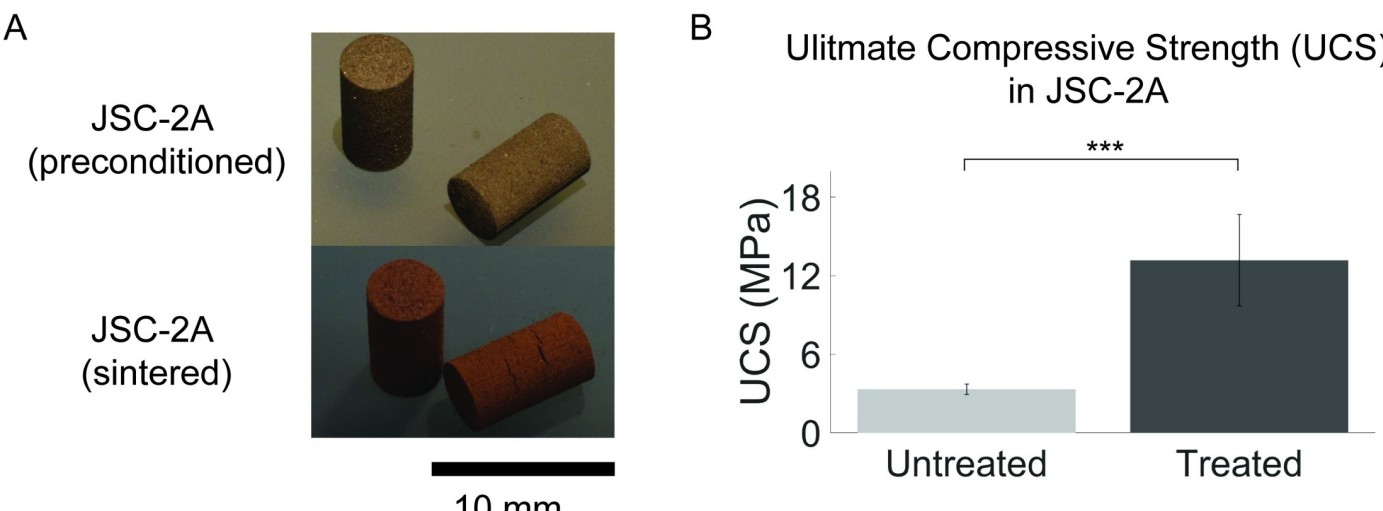

**Fig 6. Lithography-based ceramic manufacturing (LCM) prints of JSC-2A regolith-based materials.** (A) Lithography-based ceramic manufacturing of the regolith simulant JSC-2A (B) The ultimate compressive strength (UCS) of untreated (n = 9) and the bacterially treated and magnetically extracted material (n = 5). Error bars are displayed as standard deviation (Student's t-test: N.s. > 0.05; * ≤ 0.05; ** ≤ 0.01; *** ≤ 0.001).

generate locally increased pH values at the cell membrane, which is advantageous for the precipitation of magnetite [22]. First, the growth behavior and toxicity of different regolith types and concentrations to *S. oneidensis* were analysed. The growth kinetics were not negatively influenced by the presence of differing regolith simulant types or concentrations, which makes this bacterium an interesting candidate for ISRU operations.

Secondly, to better understand the interaction with the regolith and quantify *S. oneidensis* capability for ISRU, the conversion rate from Fe(III) to Fe(II) in all three tested regolith simulants was observed. Aerobic bacterial treatment of the Lunar regolith simulants EAC-1 and JSC-2A samples showed a 1.9-fold and 1.3-fold increase in dissolved Fe(II)(aq) respectively, compared to the higher 3.7-fold increase observed in bacterially treated JSC-Mars1. The difference can be explained by the fact that the two Lunar regolith simulants JSC-2A and EAC-1 consist primarily of Fe(II)-containing minerals, making them a less interesting target for bacterial treatment with *S. oneidensis* than the Mars regolith simulant JSC-Mars1, which overall has a higher iron concentration and primarily contains Fe(III). The increase in Fe(II)(aq) for JSC-2A and EAC-1 could perhaps be explained due to surface oxidation during their storage leading to increased levels of Fe(III). None of the regolith simulants was stored in an oxygen-free environment, making the outer surfaces of the materials particularly vulnerable to atmospheric oxidation. Compared to the aerobic experiments, the anaerobic bacterially induced reduction of Fe(III) in JSC-Mars1 with TSB as growth medium resulted in an even further improved 5.8-fold increase of Fe(II)(aq) after 72h. Interestingly, the Fe(II)(aq) concentration of the bacterial sample was seen to decrease after 168h. The enhanced reduction and high concentration of reduced iron under anaerobic conditions could potentially indicate a faster precipitation of the produced magnetite, making the iron unavailable for the colorimetric assay at later time-points. Anaerobic bacterial reduction of JSC-Mars1 in the defined minimal medium produced a similar result, where the Fe(II) concentration reached a maximum increase of 3.1-fold at 72 hours. The overall lower values of iron reduction observed in minimal medium compared to TSB medium might be caused by lower concentration of Fe(III) ions in the defined medium or light effects of starvation compared to the rich TSB medium.

Next, the quantity of extractable material and the total concentration of the extracts was analyzed in all three regolith types to better understand the implications for biological ISRU. The amount of magnetically extracted material increased significantly by 1.7-fold in JSC-2A after 168 hours of bacterial treatment. The magnetic extraction of EAC-1 showed significant increases in the bacterially treated sample. However, contrasting trends were observed between the measured Fe(II)(aq) concentrations (JSC-2A < EAC1) and the amount of magnetically extracted material (EAC-1 < JSC-2A) in bacterially treated samples. These results may indicate that not only the reduction, but also other bacterial functions (bioleaching, local pH) play an important role in the methodology.

The highest weight increase of magnetically extracted material (2.5-fold) and Fe(II)$_{(aq)}$ concentration (3.7-fold) under aerobic conditions was measured in bacterially treated JSC-Mars1 samples. This result was likely due to differences in starting Fe(III) concentrations among the various regolith simulants. The improved reduction and extraction in anaerobic conditions is likely due to the absence of oxygen as an alternative electron acceptor. In this environment, all electrons produced during the incubation with *S. oneidensis* are donated to the metal oxides of the regolith simulants as end terminal acceptor. Nevertheless, the maxima of extracted material of all three conditions in which JSC-Mars1 was bacterially treated (aerobically in TSB, anaerobically in TSB and minimal medium) do not significantly differ from each other. Therefore, reduction in the minimal medium under anaerobic conditions is the optimal candidate for space exploration, where the transport of additional nutrients or oxygen is a high burden on its feasibility.

Our hypothesis is that *S. oneidensis* primarily reduces the surface of the regolith simulant particles [38]. It is likely that bacterial reduction of Fe(III) to Fe(II) results in the appearance of more magnetic areas on the surface of regolith particles, so that an increased number of particles can be magnetically extracted. JSC-Mars1 has both a higher iron concentration as well as a higher Fe(III):Fe(II) ratio, presumably leading to an increased extraction yield after applying our methodology. Moreover, the JSC-Mars1 regolith simulant is expected to have a composition similar to many potential landing sites on Mars.

Finally, the extracted material was tested for 3D printing applications. The increase in iron and decrease in silicon concentration after the bacterial treatment and magnetic extractions are essential quality factors for 3D printing applications. The ultimate compressive strength was influenced by this factor, such that the bacterially and magnetically extracted 3D printed samples could withstand a pressure four times higher compared to the untreated ones. Additional 3D printing experiments coupled to material tests will lead to even greater understanding of these properties.

The methodology presented here shows the extraction of iron-rich material from different regolith simulants for 3D printing applications with improved mechanical properties compared to raw regolith. The building of sustainable habitats or replacement parts such as screws, airlocks and antennas will be all within the realm of possibility. This technology will enhance the *in-situ* resource utilization opportunities of humans using microorganisms and will, therefore, pave the way for future space exploration and Mars colonization.

# Materials & methods

## Bacterial growth conditions

*Shewanella oneidensis* MR-1 (ATCC® 700550™) was inoculated into sterilized 32 g/L Tryptic Soy Broth (TSB) media and planktonically grown overnight under aerobic or anaerobic conditions at 30˚C under continuous shaking (180–250 rpm) to an O.D.$_{600}$ of 0.5.

### Regolith pretreatment

All regolith simulant samples were sieved down to a maximum diameter of 63 μm. Only the particles smaller than 63 μm were used for further experiments.

### Preparation of defined minimal medium

The defined minimal medium was prepared by mixing 100 mM NaCl, 50 mM sodium 4-(2-hydroxyethyl)-1-piperazineethanesulphonic acid (HEPES), 7.5 mM NaOH, 16 mM NH4Cl, 1.3 mM KCl, 4.3 mM NaH2PO4·2H2O and 10 mL/L trace mineral supplement (ATCC® MD-TMS™) [39]. The defined minimal medium was autoclaved, after which it was supplemented with 10 mL/L vitamin solution (ATCC® MD-VS™) [40], 20 mg/L of L-arginine hydrochloride, 20 mg/L L-glutamate, 20 mg/L L-serine, and 20 mM lactate [41].

### Sample preparation

Regolith simulant samples (EAC-1, JSC-2A, JSC-Mars1) were sterilized via autoclaving at 121˚C and without the use of additional chemicals. The samples were diluted to the desired concentration in TSB or defined minimal medium. The aerobic samples were prepared in 15 mL Falcon tubes. The anaerobic samples were prepared in anaerobic culture tubes (Sigma Aldrich) and flushed 3x with 100% nitrogen gas to deplete the oxygen present in the solutions. *Shewanella oneidensis* overnight cultures were added to the samples to a final O.D.$_{600}$ of 0.05. The anaerobic samples were sealed with sterilized nontoxic chlorobutyl stoppers and aluminum lids (Sigma Aldrich).

### Toxicity of regolith simulants

The toxicity study was performed on aerobic and anaerobic samples containing regolith diluted in TSB at different concentrations (0 g/L, 0.1 g/L, 1 g/L, 10g/L, 100 g/L). The bacterial growth in the samples was observed in a 96-well plate via optical density (O.D.) measurements in a plate reader (Tecan infinite M200 pro microplate reader) at a wavelength of 600 nm, during incubation at 30˚C with continuous shaking (250 rpm) under aerobic or anaerobic conditions. Absorbance was measured every 5 minutes for 48 hours. The samples were measured in triplicate.

In parallel, the same samples were used to measure colony forming units (CFU). A logarithmic dilution curve was prepared of every sample to a $10^{-7}$ dilution and then plated out in triplicate onto LB agar plates at the 0, 24, and 48-hour timepoints. Dilution was carried out using phosphate-buffered saline (PBS) solution. Samples were grown for 24 hours at room temperature (20–22˚C), then individual colonies were counted.

### Small-scale magnetic extraction

The magnetic extraction set-up consisted of three 3 mL cuvettes (Sigma Aldrich) and a neodymium magnet (60x10x3 mm, coated with Ni-Cu-Ni, MIKEDE) (Fig 3A). The bottom half of the magnet was tightly wrapped in two layers of aluminum foil and two layers of Parafilm. The design allowed for an easy removal and insertion of the magnet. The first cuvette was filled with 1 mL of the sample, and the second and third were filled with MilliQ water. The magnet with cover was inserted into the first cuvette twice. Magnetic material was attracted to the magnet's cover, and the covered magnet with associated magnetic material was transferred to the second cuvette. The magnet was carefully removed from the cover to allow the magnetic material to sink into the cuvette. Then the magnet was inserted again to attract only magnetic material, which was transferred into a third pre weighted cuvette. To suspend the magnetic

material within the MilliQ water, an additional neodymium magnet was placed underneath the cuvette. The liquid, which was not attracted by the magnet was decanted, and the cuvettes were weighted after they had dried. The original weight of each individual cuvette was subtracted from the total weight after addition of extracted material.

Aerobic and anaerobic Shewanella oneidensis samples with 10 g/L regolith diluted in TSB (or in defined minimal medium for anaerobic samples) were incubated for 168 hours. The weight of the extracted material was determined at the 0 and 168-hour timepoints for the aerobic samples, and at the 0, 72, and 168-hour timepoints for the anaerobic samples.

## Fe(II)$_{(aq)}$ concentration determination

The aqueous ferrous iron concentrations were determined based on a colorimetric method [42], which is a modification of the Phenanthroline method [35]. By dissolving 0.6 grams sodium fluoride (Sigma-Aldrich, 99%) in 28 mL MilliQ water and 0.57 mL sulfuric acid (99.999%), a complexing reagent was formed. An o-Phenanthroline solution was prepared mixing 30 μL of hydrochloric acid (37%) with 7 mL MilliQ water and dissolving 0.2 grams 1,10-Phenanthroline monohydrate (Sigma-Aldrich, 99%) in it. The complexing reagent was shaken, and 10 mL of MilliQ water was added.

An acetate buffer was prepared by mixing 3 mL MilliQ water with 12 mL acetic acid (50 mM) and dissolving 5 grams ammonium acetate (Biosolve B.V.) into it. The acetate buffer was shaken, and MilliQ water was added up to 20 mL. The reaction reagent was prepared by mixing 5 mL of the o-Phenanthroline solution with 5 mL of the acetate buffer. To determine the aqueous ferrous iron concentration of a specific sample, 0.1 mL of the sample was pipetted into a 5 mL microtube (Eppendorf). Next, 1 mL of the complexing fluoride reagent was added, the solution was shaken, and 0.4 mL of the reaction reagent was pipetted into it. Samples were agitated, then MilliQ water was added to the samples up to 2.5 mL. A volume of 200 μL of the sample was pipetted in triplicate into a 96-well plate (SARSTEDT). Samples were incubated for five minutes, then the absorbance of the 96-well plate was measured at 510 nm (Tecan infinite M200 pro microplate reader).

A Fe(II) standard curve was prepared by assaying ammonium iron(II) sulfate hexahydrate dissolved in TSB to a final concentration of 0, 50, 100, 150, or 200 ppm. A second standard curve was prepared by assaying samples containing the same Fe(II) concentrations as well as 100 ppm iron(III) citrate.

Aerobic and anaerobic Shewanella oneidensis samples with 10 g/L regolith diluted in TSB (or in defined minimal medium for anaerobic samples) were incubated for 168 hours. Iron concentration was determined at the 0, 48, 72, and 168-hour timepoints.

## Large-scale magnetic extraction

Large-scale magnetic extractions were performed by preparing a 10 g/L solution of previously sieved (<63 μm) EAC-1, JSC-2A, or JSC-Mars1 regolith simulants with TSB medium. An overnight culture of Shewanella was added to an O.D.$_{600}$ of 0.05 and grown for 168h at 30°C with shaking at 120 rpm. Neodymium magnets (60x10x3 mm, coated with Ni-Cu-Ni, MIKEDE) were wrapped in two layers of aluminum foil covering the ends of the magnets, after which the foil-tipped magnets were wrapped completely in a layer of Parafilm. After 168-hour incubation, the wrapped magnet was placed into the solution, the flask was swirled five times, the material was allowed to settle for 30 seconds, and the magnet was extracted again by using rubber tweezers and additional neodymium magnets. The extracted magnet was washed in a 50 mL Falcon tube containing MilliQ water. Scissors were used to cut open the middle region of the Parafilm magnet-cover, the magnet was removed from the cover, and

the cover was rinsed into an empty Petri dish. This process was repeated until the amount of extracted material per magnet was so little that continuation would not result in an effectively higher yield. The non-magnetic material was collected by rinsing the remaining material out of the flask into an additional Petri dish.

## X-ray fluorescence measurements (XRF)

A pressed powder tablet was prepared by adding 0.25 g Boreox (FLUXANA) to 1 g of the tested regolith simulant (untreated and treated JSC-2A, JSC-Mars1, or EAC-1). The mixture was milled using a malachite mortar until a uniform mixture was achieved. About 5 g of Boreox was added to a hollow metal cylinder and pressed on a hydraulic press up to 10 kPa/cm$^2$ (P/O/Weber Laborpresstechnik). The mixture of regolith and Boreox was added to the metal cylinder and pressed up to 250 kPa/cm$^2$. The pressed tablet was analyzed with an X-ray fluorescence spectrometer (Axios Max WD-XRF, Malvern Panalytical Ltd). Analysis of the XRF data was performed with SuperQ5.0i/Omnian software.

## X-ray diffraction measurements (XRD)

Approximately 100 mg of regolith simulant was equally distributed over a silicon crystal sample holder (Si510 zero-background wafer). XRD measurements were taken using a diffractometer (D8 Advance, Bruker-AXS) with Bragg-Brentano geometry and a Lynxeye position-sensitive detector. Cu Kα radiation was used at under 45 kV and 40 mA, with a scatter screen height of 5 mm. The sample was scanned with an X-ray beam varying from 8˚–110˚, with a step size of 0.021˚ * 2θ (in which θ is the angle between the incident ray and the surface of the sample) and a counting time of 1 second per step. Analysis of the XRD data was performed with Bruker—AXS software DiffracSuite.EVA v 5.0.

## X-ray photoelectron spectrometry (XPS)

Samples were analyzed with X-ray photoelectron spectroscopy (XPS) on a ThermoFisher K-Alpha system using Al Kα radiation with a photon energy of 1486.7 eV. Powder samples were immobilized onto a copper tape (Plano GmbH, G3397) and were loaded into the XPS chamber without further purification. Iron high-resolution XPS spectra were acquired using a spot size of 400 μm, 50 eV pass energy, and 0.1 step size, averaging 50 scans from 705 eV to 740 eV with charge neutralizing. The peaks were calibrated for the C 1s peak at 285 eV. The background was subtracted using the "smart" function of the ThermoFisher Advantaged software.

## Lithography-based ceramic printing

The regolith simulant JSC-2A was ball-milled and mixed with 25.8 wt% of a photocurable organic binder (Lithoz) to prepare a printing feedstock. 3D printed pre-sintered green parts were produced on a CeraFab 7500 using vat photopolymerization. A light engine (based on LEDs) with a digital micromirror device was used to selectively harden the previously prepared regolith simulant containing feedstock layer by layer into desired shape based on a given STL file. The 3D printed green parts were subsequently dried and sintered for 2 hours at 1050˚C and 1100˚C for the untreated and treated samples, respectively.

## Ultimate compressive strength of 3D prints

Compressive tests were performed to determine the ultimate compressive strength of the 3D printed materials (Instron ElectroPuls E10000 Linear-Torsion) under environmental

conditions of 20˚C and a relative humidity of approximately 50%. DIN 51104 standards were followed with two exceptions: no intermediate plate was used between the instrument and the samples, and the specimens were not fully compliant to the standard since the ends of the 3D prints were not perfectly flat. A constant deformation rate of 0.5 mm/min was used. The ultimate compressive strength was calculated using the following formula: $\sigma = F^* / A_0$, where $\sigma$ is the ultimate compressive strength, $F^*$ is the force applied to the sample just before it cracks, and $A_0$ is the initial area of the sample.

## Supporting information

**S1 Data.**
(ZIP)

## Acknowledgments

Our thanks to Florian Ertl for the 3D printing work performed and to Hubertus J. E. Beaumont for providing access to their laboratories.

## Author Contributions

**Conceptualization:** Stan J. J. Brouns, Anne S. Meyer.

**Funding acquisition:** Stan J. J. Brouns, Anne S. Meyer.

**Investigation:** Sofie M. Castelein, Tom F. Aarts, Juergen Schleppi, Dominik Benz, Maude Marechal, Advenit Makaya, Martin Schwentenwein, Benjamin A. E. Lehner.

**Methodology:** Juergen Schleppi, Ruud Hendrikx, Dominik Benz, Maude Marechal, Martin Schwentenwein.

**Resources:** Amarante J. Böttger.

**Supervision:** Anne S. Meyer, Benjamin A. E. Lehner.

**Validation:** Advenit Makaya.

**Writing – original draft:** Sofie M. Castelein, Tom F. Aarts, Benjamin A. E. Lehner.

**Writing – review & editing:** Tom F. Aarts, Anne S. Meyer, Benjamin A. E. Lehner.

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
