## [Decision Letter · Decision Letter 0]

15 Jan 2021

PONE-D-20-33186

Iron can be microbially extracted from Lunar and Martian regolith simulants and 3D printed into tough structural materials

PLOS ONE

Dear Dr. Lehner,

Thank you for submitting your manuscript to PLOS ONE. After careful consideration, we feel that it has merit but does not fully meet PLOS ONE’s publication criteria as it currently stands. Therefore, we invite you to submit a revised version of the manuscript that addresses the points raised during the review process.

We look forward to receiving your revised manuscript.

Kind regards,

Amitava Mukherjee, ME, Ph.D.

Academic Editor

PLOS ONE

Journal Requirements:

"No competing interests are declared"

We note that one or more of the authors are employed by a commercial company: Lithoz GmbH.

2.1. Please provide an amended Funding Statement declaring this commercial affiliation, as well as a statement regarding the Role of Funders in your study. If the funding organization did not play a role in the study design, data collection and analysis, decision to publish, or preparation of the manuscript and only provided financial support in the form of authors' salaries and/or research materials, please review your statements relating to the author contributions, and ensure you have specifically and accurately indicated the role(s) that these authors had in your study. You can update author roles in the Author Contributions section of the online submission form.

2.2. Please also provide an updated Competing Interests Statement declaring this commercial affiliation along with any other relevant declarations relating to employment, consultancy, patents, products in development, or marketed products, etc.  

Reviewers' comments:

Reviewer's Responses to Questions

**Comments to the Author**

1. Is the manuscript technically sound, and do the data support the conclusions?

Reviewer #1: Yes

Reviewer #2: Yes

2. Has the statistical analysis been performed appropriately and rigorously? 

Reviewer #1: Yes

Reviewer #2: Yes

3. Have the authors made all data underlying the findings in their manuscript fully available?

Reviewer #1: Yes

Reviewer #2: Yes

4. Is the manuscript presented in an intelligible fashion and written in standard English?

Reviewer #1: Yes

Reviewer #2: Yes

5. Review Comments to the Author

Reviewer #1: SI enjoyed reading this report and found can the potential this work has to contribute to ISRU. From the perspective as a reviewer the report found that Shewanella has potential as a practical tool for biomining and extraction of useful materials, in this case iron from Martian regolith, and even improves the quality of materials for use. This is particularly useful given that bacteria are less cumbersome to bring to Mars than heavy equipment and the work here provides a proof of concept level of work contributing to the preliminary foundation that this technology can be applied to space exploration. I thought overall the authors did well at summarizing and setting up the topic in abstract and introduction providing adequate context for why the work is significant, though there are some specific points here that may need clarification or correction (see below) overall the work is well set up in this aspect. There are some areas of concern that I do recommend for revision and these are included in the attachment.

Reviewer #2: The manuscript reports on a novel approach to extract Iron from Lunar and Martian simulant material using Shewanella oneidensis bacteriun. An additional bonus is included by demonstrating the increased material strength properties of subsequently 3d printed test articles. As a demonstration of biology driven ISRU process, this work is highly novel and the results would be of great interest to those working on novel ISRU approaches.

6. PLOS authors have the option to publish the peer review history of their article (what does this mean?). If published, this will include your full peer review and any attached files.

Reviewer #1: No

Reviewer #2: No

---

## [Author Response · Author response to Decision Letter 0]

9 Mar 2021

Response to the reviewers comments can be found in the dedicated file attached. 

Response to the editors comments: 

1) We used the given templates to adopt the manuscript to PLOS ONE's style

2) We agree to the update of the funding and competing interest section as described above.

3) All data was added as supporting information

---

## [Decision Letter · Decision Letter 1]

29 Mar 2021

Iron can be microbially extracted from Lunar and Martian regolith simulants and 3D printed into tough structural materials

PONE-D-20-33186R1

Dear Dr. Lehner,

We’re pleased to inform you that your manuscript has been judged scientifically suitable for publication and will be formally accepted for publication once it meets all outstanding technical requirements.

Kind regards,

Amitava Mukherjee, ME, Ph.D.

Academic Editor

PLOS ONE

Additional Editor Comments (optional):

Reviewers' comments:

Reviewer's Responses to Questions

**Comments to the Author**

1. If the authors have adequately addressed your comments raised in a previous round of review and you feel that this manuscript is now acceptable for publication, you may indicate that here to bypass the “Comments to the Author” section, enter your conflict of interest statement in the “Confidential to Editor” section, and submit your "Accept" recommendation.

Reviewer #1: All comments have been addressed

2. Is the manuscript technically sound, and do the data support the conclusions?

Reviewer #1: Yes

3. Has the statistical analysis been performed appropriately and rigorously? 

Reviewer #1: Yes

4. Have the authors made all data underlying the findings in their manuscript fully available?

Reviewer #1: Yes

5. Is the manuscript presented in an intelligible fashion and written in standard English?

Reviewer #1: (No Response)

6. Review Comments to the Author

Reviewer #1: (No Response)

7. PLOS authors have the option to publish the peer review history of their article (what does this mean?). If published, this will include your full peer review and any attached files.

Reviewer #1: No

---

## [Editor Report · Acceptance letter]

6 Apr 2021

PONE-D-20-33186R1 

Iron can be microbially extracted from Lunar and Martian regolith simulants and 3D printed into tough structural materials 

Dear Dr. Lehner:

I'm pleased to inform you that your manuscript has been deemed suitable for publication in PLOS ONE. Congratulations! Your manuscript is now with our production department. 

Kind regards, 

on behalf of

Professor Dr. Amitava Mukherjee 

Academic Editor

PLOS ONE